# Nanocomposites Materials of PLA Reinforced with Nanoclays Using a Masterbatch Technology: A Study of the Mechanical Performance and Its Sustainability

**DOI:** 10.3390/polym13132133

**Published:** 2021-06-29

**Authors:** Helena Oliver-Ortega, Josep Tresserras, Fernando Julian, Manel Alcalà, Alba Bala, Francesc Xavier Espinach, José Alberto Méndez

**Affiliations:** 1Group LEPAMAP-PRODIS, Department of Chemical Engineering, University of Girona, C/M. Aurèlia Capmany, 61, 17003 Girona, Spain; helena.oliver@udg.edu (H.O.-O.); jose.tresserras@udg.edu (J.T.); fernando.julian@udg.edu (F.J.); manuel.alcala@udg.edu (M.A.); francisco.espinach@udg.edu (F.X.E.); 2UNESCO Chair in Life Cycle and Climate Change ESCI-UPF, Universitat Pompeu Fabra, Passeig Pujades 1, 08003 Barcelona, Spain; alba.bala@esci.upf.edu

**Keywords:** nanocomposites, nanoreinforcement, mechanical performance, sustainability

## Abstract

Packaging consumes around 40% of the total plastic production. One of the most important fields with high requirements is food packaging. Food packaging products have been commonly produced with petrol polymers, but due to environmental concerns, the market is being moved to biopolymers. Poly (lactic acid) (PLA) is the most promising biopolymer, as it is bio-based and biodegradable, and it is well established in the market. Nonetheless, its barrier properties need to be enhanced to be competitive with other polymers such as polyethylene terephthalate (PET). Nanoclays improve the barrier properties of polymeric materials if correct dispersion and exfoliation are obtained. Thus, it marks a milestone to obtain an appropriate dispersion. A predispersed methodology is proposed as a compounding process to improve the dispersion of these composites instead of common melt procedures. Afterwards, the effect of the polarity of the matrix was analyzing using polar and surface modified nanoclays with contents ranging from 2 to 8% *w*/*w*. The results showed the suitability of the predispersed and concentrated compound, technically named masterbatch, to obtain intercalated structures and the higher dispersion of polar nanoclays. Finally, the mechanical performance and sustainability of the prepared materials were simulated in a food tray, showing the best assessment of these materials and their lower fingerprint.

## 1. Introduction

The use of plastics materials has increased every day since the 1950s and nowadays has become essentials in our life. The advantages of these materials are clear: low cost, low weight, easy methodologies for its transformation, etc. However, the origin and recyclability of a high part of those plastics are unsustainable [1]. The increased use of polymers was produced as an alternative to non-renewable and scarce materials, and despite the non-renewable behavior of most of them, during the fast rise of its production and consumption, petroleum was considered abundant. Nonetheless, it is known and well established that petroleum is finite, and society has to shift to more sustainable and bio-based materials [2,3].

Another problem with common plastics is their accumulation and presence in the environment [4]. Generally, common plastics can be efficiently recycled in mechanical or chemical forms. In Europe, the recyclability tax in 2019 was 32.5% of the total plastics collected post-consumption. The tax is lower than that found in energy recovered (42.6%) and closer to the landfilled (24.6%), although a 19% increment has been obtained in the recyclability tax for the last 12 years [5]. Moreover, the recyclability tax is low and landfill is a common practice in our society. One of the fields that shows high plastic consumption of (usually) single-use plastics, which contribute actively to the increment of its accumulation, is packaging. This sector consumes around 40% of the plastic production in Europe and just 42% of the recollected residues (17.8 Mt) are recycled. A drastic change in behavior and ethics is necessary for this field and society. In that sense, one of the most studied options is the use of renewable and biodegradable plastics. These plastics can maintain market requirements while avoiding the accumulation problem as they are biodegradable and could be compostable. Moreover, in some cases, these plastics can be recycled so the lifespan is not restricted to one single use. Among these renewable and biodegradable plastics, poly(lactic acid) (PLA) has become one of the most promising polymers [6,7].

The main requirements in packaging to replace common materials used in the field with sustainable ones are: to be processed with the same or similar technologies and equipment; to maintain the mechanical properties and; and to keep the same or improved barrier properties. PLA has similar or better mechanical properties than the main plastics used in packaging—polyethylene (PE), polypropylene (PP), and polyethylene terephthalate (PET)—and can be processed with the same equipment [8]. Nonetheless, the PLA barrier properties are adequate but need improving to expand their use in the food packaging field. These properties can be improved by using a multilayer film structure or by its reinforcement with some particles such as nanoclays [9].

Composites materials reinforced with nanoclays improve their barrier properties if the nanoclays are correctly dispersed in the matrix [10]. These particles produce difficulty in the gas permeability through the material, slowing down its passage [11,12,13]. However, nanomaterials are not easy to disperse, moreover with polar behavior, in apolar matrices. Good to moderate results have been obtained in nylons and starch using organically modified and unmodified nanoclays resulting in intercalated and exfoliated materials [14,15,16,17]. However, both matrices have a higher polarity than PLA, which facilitates their dispersion in the matrix. However, some aggregations have been observed in the samples, mainly depending on the production methodology [12]. A normal way to improve the dispersion through the matrix is the use of organomodified nanocomposites [18]. The use of these kinds of modifications in the nanoparticles could avoid the loss in the mechanical properties of the nanocomposties [19,20]. Nevertheless, such materials could present some problems as some of them cannot be used in food packaging material due to their probable human toxicity and the increased price regarding the polar ones directly obtained from nature.

The cheapest form to obtain composites and nanocomposites is by its direct mixing with the matrix in extrusion equipment, which could produce it in a continuous form or in a batch methodology, obtaining the blends discontinuously. Nonetheless, the correct composting of nanoclays is difficult as they are in a dust form and their additivation is complicated and the correct content cannot be ensured. Moreover, the exfoliation is not easily obtained in that form and has a large dependency on the speed and time of the material extrusion. However, this parameter has also a high impact on the degradation of the material during the process and could induce lower properties due to the degradation. To avoid this, a premixing procedure, technically named masterbatch, is evaluated and compared to direct mixtures of PLA and nanoclays. The masterbatch procedure which is used in polymer colorants, for example, consists of the production of a small quantity of concentrated polymer-reinforced nanocomposite, which will be diluted in neat polymer to obtain the correct reinforcement composition. This methodology avoids the errors produced in the direct mixing, facilitates the additivation process, and could lead to better dispersion of the nanoparticles due to the two mixing processes [21]. In our study, the results showed that the masterbatch methodology maintained or improved the dispersion of the PLA nanocomposites while facilitating the mixture process. Afterwards, the use of polar clays was compared to organically modified, for the production of PLA nanocomposites with 2,4 and 8% *w*/*w* of reinforcement. The comparison was performed to test the effect of the affinity of the polymer matrix and the nanoclay reinforcement in the new dispersion methodology. Finally, the modeling of a food tray with PLA nanocomposites and common plastics of the market was performed to establish the suitability of these materials to replace the common ones in terms of mechanical performance and sustainability.

## 2. Materials and Methods

### 2.1. Materials

PLA Ingeos Biopolymer 3251D (Nature Works, Naarden, The Netherlands) was used as a polymer matrix to produce the nanocomposite materials. The nanoclays used as reinforcement were hydrophilic bentonite and surface modified nanoclays containing trimethyl stearyl ammonium, both supplied by Sigma Aldrich (Madrid, Spain). The sizes of the provided particles were 6 and 10 microns and the bulk densities were 779 kg·m^−3^ and 336 kg·m^−3^ for the unmodified and modified nanoclays, respectively.

### 2.2. Methods

#### 2.2.1. Compounding Process

To achieve correct compounding in nanomaterials is a milestone. To assess the better compounding procedure, a masterbatch methodology was studied instead of the direct one. Initially, 4% wt-reinforced nanocomposites with unmodified nanoclays were produced by the direct and the masterbatch methodology to test the effectiveness of the masterbatch method. Nanocomposites were produced and mechanically tested. The first methodology is to produce directly the nanocomposite by mixing the nanoclay and the PLA in an internal mixer (Brabender^®^ plastograph mixing machine, Duisburg, Germany). First, the polymer is melted in the mixing chamber at 195 °C and 45 rpm. Then, the nanoclays are added slowly in the chamber to avoid the maximum loss possible during the process. The mixing is done once the torque is stabilized and the total mixing time is about 5 min. The second tested methodology, the masterbatch one, started with the production of a concentrated blend (22% wt nanoclays) and dilute the blend later with fresh polymer in a high-intensity mixer (Gelimat Kinetic Mixer, Ramsey, NJ, USA) to obtain the preferred reinforcement composition. The concentrated blend was produced in a Brabender^®^ plastograph with the process mentioned above. Then, the obtained blend was mixed with PLA in a Gelimat Kinetic Mixer. The polymer and the masterbatch were added at a low speed (300 rpm) that increases up to 2500 rpm to perform the mixing. Once the temperature of the chamber achieves 200 °C, the material is discharged, cooled down, and milled to obtain the adequate pellet size for the transformation process. The milling process was also used for the samples from the direct mixing.

Samples for mechanical testing were obtained by injection molding using an Aurburg 220 M 350–90 U equipment (Aurburg, Loßburg, Germany). The temperature profile was 180–190–200–210 °C and the pressures ranged from 300 to 350 bars depending on the nanoclay content.

#### 2.2.2. Mechanical Characterization

Samples were mechanically characterized to tensile properties in a DTC-10 Universal testing machine (IDMtest, New York, NY, USA) equipped with a 5 kN load cell. The Young’s modulus was obtained using an extensometer. Before the tests, samples were conditioned at 23 °C and 50% of relative humidity for 48 h in a Dycometal conditioning chamber following ASTM D618 standards. Statistical analyses of the samples were carried on with an ANOVA test (95% of confidence) using Origin 2019 software.

#### 2.2.3. Thermal Behaviour

Thermal properties were studied through Thermogravimetric Analysis (TGA) and Differential Scanning Calorimeter (DSC) techniques. A Mettler Toledo SDTA 851 thermobalance (Mettler Toledo, L’Hospitalet de Llobregat, Spain) was used for the TGA working in a range of temperatures from 30 to 700 °C, a heating rate of 10 °C/min in an inert atmosphere (N_2_, 0 mL/min). DSC tests were performed in a Mettler Toledo DSC822e calorimeter (Mettler Toledo, L’Hospitalet de Llobregat, Spain) following ASTM E 1269.01 standard specifications. The tests were performed in an inert atmosphere (N_2_) with a constant flow of 40 mL/min. The heating rate was 10 °C/min and the thermal range was 30 to 200 °C for all the samples. Thermal history was firstly erased in all the samples using the same conditions.

#### 2.2.4. X-ray Diffraction (XRD)

Samples structure and nanoclay exfoliation and/or intercalation were analysed using X-ray Diffraction (XRD) in a D8 QUEST ECO (Bruker, Madrid, Spain) with a Cu-Kα radiation (λ = 0.15406 nm). Data were collected on the 2 θ range from 1.2° to 30° operating at 40 KV and 40 mA. The d-spacing of the different planes analysed were calculated from Bragg’s Law:(1)λ=2dsin θ

#### 2.2.5. Scanning Electron Microscopy (SEM)

A Zeiss DSM 960A (Carl Zeiss Iberia, Madrid, Spain) Scanning electron microscope (SEM) was used to obtain micrographs from the fracture of tensile samples. The samples were gold-coated before observation.

#### 2.2.6. Sample Modeling and Evaluation

A digital mock-up was built in similarity with a commercial sample available in supermarkets to model and evaluate the performance of the PLA nanocomposites in comparison with commercial ones. The model was simplified to ease its meshing and analysis. Fillet radius lower than 1 mm were avoided and ribs were excluded from the digital mock-up. The model was made as a solid using the SolidWorks CAD software by Dassault Systemes (France). The finite element analysis was performed in the same software with the advanced simulation analysis package. The simulation included the usual steps for such analysis; definition of the material properties (Young’s modulus, Poison’s ratio, and tensile strength), placing movement restrictions and loads, meshing the model, launching the analysis, and review of the results. The solid was meshed using hexahedron elements and the mesh was refined until considered correct avoiding elements with aspect ratios larger than 3.

A preliminary life cycle analysis (LCA) was performed using the LCA module included in SolidWorks 2020 software. Unavailable data for PLA composites materials were obtained in the literature. In all the cases, the materials were assumed to be produced, manufactured, and recycled, or landfilled in Europe.

## 3. Results and Discussion

### 3.1. Evaluation of the Compounding Process

The effect of the compounding methodology was evaluated from its mechanical properties. It is well accepted in the literature that tensile and flexural strength and modulus are highly dependent on reinforcement dispersion, orientation, content, and chemical structure [22]. Reinforcement aggregates produce a discontinuity in the polymer matrix and difficult the stress transmission and material’s deformation. Thus, poor interaction between the polymer matrix and the reinforcement can lead to poor mechanical properties [23]. Nonetheless, a thermal degradation due to the processing method could also lead to a reduction of the mechanical properties.

PLA nanocomposites reinforced with 4% of polar nanoclays (PLA+4%N) were produced with the masterbatch and the direct mixing methodologies. The polar nanoclays were chosen as reference reinforcement as the highly polar behaviour of this material will add another difficulty in the mixing process. The results from the mechanical testing are shown in Table 1.

Young’s modulus showed an increment in PLA nanocomposites regarding the polymer matrix using both methodologies. The values were higher for the masterbatch methodology, but the slight difference between both compounding methods seems to not be enough to be considered representative. However, the increase regarding the polymer matrix seems to indicate that the 4% *w*/*w* can be adequately dispersed in the PLA matrix besides the methodology used. Tensile resistance showed almost the same value as PLA, a little bit lower in the case of the direct compounding methodology. In that case, any reinforcement effect is observed by the nanoclays in the PLA matrix as the tensile strength is maintained. A similar effect is observed in the deformation. A small increment is obtained in PLA+4%N Masterbatch, but the difference is too low to be considered significant. The similar values seem to indicate a poor interaction with the filler and the presence of an intercalated structure. This type of structure was expected as different studies showed that just small quantities of nanoclays, and usually organic modified ones, obtained the fully exfoliated nanoclay layers by melt compounding methods [12,24,25].

The XRD of the composites materials was performed in order to determine the intercalation effect of the samples. Figure 1 shows the obtained results.

Nanoclay materials used showed a peak corresponding to the plane 001 around 2θ ≈ 6.9°. However, the diffractograph of the composite materials showed a small peak corresponding to this plane at 2θ ≈ 6.0°. The shift in the d-spacing from 1.27 nm of the nanoclays, in the range obtained in the literature [24,25], to a distance of 1.47 nm and 1.46 nm for the masterbatch and directly prepared composites, indicates some degree of intercalation of the nanoclays in the nanocomposites [26]. The other broad peak around 16° in nanocomposites corresponds to the PLA amorphous and overlaps with the nanoclay peak around 20°. Moreover, the intensity of the nanoclay peak in the masterbatch methodology and also the broader and less intense peak of the PLA in that methodology could lead to considering better intercalation and some exfoliation of the nanoclays in the masterbatch procedure.

SEM micrographs of the tensile fractures were measured for both methodologies (Figure 2). A particle dispersed structure could be observed for both methodologies (Figure 2A,B). Nanoclay aggregates could be easily observed in both cases. This was expected as it is difficult to obtain a good dispersion of them by direct melting and in nanoclay reinforcement with a higher content than 2–3%. Better dispersions are observed in other polymers when a liquid dispersion is performed [27,28,29]. The magnified images demonstrate the poor affinity of nanoclays with the polymer matrix as void volume could be observed around the aggregates (Figure 2C–F). Moreover, at ×10,000 magnification, it is possible to observe layers of nanoclays separated from the aggregate and incorporated into the polymer, indicating some degree of intercalated structure, as was previously considered in the nanocomposites.

The results obtained from the XRD and the mechanical properties indicated a slightly better dispersion of the nanoclays by the masterbatch methodology. Such methodology facilitates the production of these kinds of nanocomposites regarding direct melting at is easy and safe to control the nanoclay addition to the polymer matrix. Moreover, it includes a two-steps processing which could produce some effect on the thermal stability of nanocomposites. Parallel, the effect of nanoclays on the thermal stability of PLA is well known [30,31,32]. The addition of nanoclays usually led to a slight increment of the degradation temperature of the matrix when a correct dispersion and exfoliation are obtained. The thermal stability was analysed through TGA (Figure 3).

The TGA results do not show any significant changes between both methodologies. One single degradation step is observed in the TGA corresponding to the polymer degradation (Figure 3). Table 2 shows the temperatures extracted from the graph. It could lead to considering no effect of the masterbatch two-steps procedure. An increment of the T_10%_ is observed in both methodologies, and also a slight displacement of the T_max_ temperature. This was expected as the mechanical results seem to indicate an adequate dispersion and the presence of some intercalated layers [12,22]. The slightly higher results of PLA+4%N Brabender could be related to the additional step of the masterbatch methodology. However, the differences do not seem to affect significantly the degradation temperatures. The residue of both samples is close to the expected as 4% of mineral content is loaded in the material.

DSC of the samples was performed to evaluate the effect of the compounding in the main transition temperatures of the matrix. Moreover, changes in these temperatures could affect the working temperatures of the materials. DSC thermographs after erasing the thermal history are shown below in Figure 4:

A displacement, around 2 °C, of the glass transition temperature (T_g_) of the PLA is observed in nanocomposites. The displacement reinforces the hypothesis of the production of an intercalated structure in nanocomposites, as these structures can inhibit the polymer mobility at low quantities of nanoclays [33,34]. Highly dispersed and exfoliated materials with good compatibility with PLA showed a plastifying effect [35]. The melting process is slightly affected by the addition of nanoclays. The same phase change is observed and a small displacement in the melting temperature from 167.5 °C to 168.8 and 169.3 °C in the direct and masterbatch methodologies, respectively. Nonetheless, the crystallinity decreased from 53.6% for PLA to 39.8 and 37.8% for the direct and masterbatch methodologies, respectively. The reduction in crystallinity is usually associated with a strong interaction between polymer and nanoclays and is generally found in organically modified nanoclays and exfoliated samples [36]. In intercalated nanocomposites, a nucleating effect is showed generally but with smaller spherulites [37]. Although the reduction of the crystallinity can indicate an exfoliated structure, the cold crystallization behavior is associated with intercalated structures. Moreover, it is in concordance with the observed in the previous assays and the SEM indicated a small degree of intercalation. The cold crystallization of PLA shows a reduction of the temperature and a broader peak for the nanocomposites, with the lower temperature obtained for the masterbatch methodology. The temperature peaks of the cold crystallization were 104.3 °C for the neat PLA, 101.4 °C for direct processing, and 98.7 °C for the masterbatch one. Thus, the reduction in crystallinity could be also related to the poor interaction of the matrix and the nano reinforcements, which was clearly observed in the SEM micrographs.

The masterbatch process was selected over the direct mixing because it produced slightly better results and facilitates the process although two mixings are required, one for the production of the concentrated masterbatch and the second one for the composite production. The acquisition of a concentrated sample led to avoid working with the nanoclays in a dust form and to be more accurate during the mixture of the phases and healthier. The results are quite representative as the Gelimat kinetic mixer is semi-industrial equipment. Moreover, it is one of the reasons that masterbatch technology has been established in colorants for plastics manufactures. On the other hand, in the direct mixing, the composite was obtained in long-time mixings (5 min) in lab-scale equipment. Industrial processes are generally done in a screw-extruder with shorter production times and probably lower dispersion results than the obtained in the lab-scale equipment.

### 3.2. Comparison between Nanoclay Type and Composition

Once the methodology was established, composite materials modified by the addition of polar nanoclays and surface modified nanoclays, referenced as PLA+X%NSF, were prepared and mechanically characterized. Table 3 shows the results obtained.

Surface-modified nanoclays are expected to have a better dispersion due to their higher affinity with the polymer. Thus, the use of the masterbatch technology could improve its dispersion. The highest tensile moduli were obtained for surface-modified nanoclays (NSF). This could be related to that better dispersion of the clays and probably a higher degree of intercalation. In the case of polar nanoclays, the increment of the nanoclay content had a lower impact than in the NSF nanoclays. Previously, an intercalated structure was considered for PLA+4%N, and a similar structure can be expected for both nanofillers although it is expected to be higher in NSF types. The reduction in deformation is also a clear indication of that intercalated structures as it is not highly reduced at low contents. In the case of 2% nanocomposites, a more exfoliated structure could be the cause of higher mechanical properties [13,36]. Nonetheless, the reduction of the tensile resistance and the deformation with the nanofiller content is reduced in polar composites in comparison with NSF. This effect can be associated with the reduction of the molecular weight of PLA observed in organic modified nanoclay-reinforced composites [30]. The addition of another thermal step in the compounding process can intensify this degradation, resulting in lower mechanical properties. Nevertheless, in terms of mechanical performance, all the composites seem to be able to compete with common plastics for food packaging such as PE, PP, and PET.

The exfoliation and intercalation of the nanoclays in the materials were also studied by XRD. The XRD patterns of polar nanoclays and NSF nanocomposites are shown in Figure 5.

A clear different XRD pattern was obtained for both types of composites. In the polar nanoclay nanocomposites, the peak around 6.0° related to the nanoclays is not clearly observed in the composite with 2% *w*/*w*. That result could be related to a highly exfoliated and intercalated material, however, the low quantity of the sample could also difficult its detection as in higher contents the intensity of the peak is also low [36]. Nonetheless, the 2θ angle is shifted to lower angles than the observed ones in nanoclays that corresponded with higher the d-spacing of that plane. Moreover, the distance is increased when the content is reduced, indicating higher intercalation in lower contents. Thus, the 2%-reinforced material could be also exfoliated.

In the NSF-reinforced nanocomposites, four peaks are clearly observed for the nanocomposites in the range of 1.2 to 10° and three peaks appear in the PLA amorphous when the nanoclay content is increased in the material. Moreover, the peaks between 1.2 to 10° are not observed in NSF, where just two peaks around 1.6 and 4.2° are observed. The nanoclays were probably separated and intercalated, so a different behaviour was observed from powder to composites. The d-spacings related to the nanoclay layers around 5° and 7° showed values of 1.8–2 nm and 1.2 nm, respectively [38,39]. In the case of the basal peak corresponding to the peak around 2°, the d-spacing is 3.6–3.7 nm [40]. The values are quite similar to those reported in the literature and the small differences observed indicate some grade of intercalation. Nonetheless, it seems the intercalation degree of these nanoclays is not so high and the improvements in the mechanical properties are more devoted to particle sizes and correct dispersion and interaction with PLA.

Different behaviour is observed in PLA from both composites. In the polar nanocomposites, the broad peak is related to a high amorphous material (the crystallinity reported in the initial form obtained from DSC indicates values ranging from 16 to 20%). In the NSF-reinforced composites, the increment of the NSF content led to the apparition of some peaks, the main one around 16.5°. The apparition of these peaks could be related to a different crystalline structure in PLA, and similar results are observed in annealed samples [41]. Nonetheless, the results obtained from the 1st melting of the DSC showed lower crystallinity values and no different phases are observed in the melting indicating different crystalline phases. Other authors proposed that these peaks could be related to degradation. Moreover, a small peak is observed also in the PLA+8%N composite that could be related to the degradation phenomena [18,42]. A degradation in the material is in concordance with the observed in the resistance and deformation behaviour in the mechanical properties.

SEM micrographs, produced of all the materials, are shown in Figure 6. Nanocomposites with a 2% reinforcement showed clear differences in the particle size, indicating a better dispersion in the NSF type. Nonetheless, small and intercalated ones seem to be also found in polar nanoclays. In the case of 4%, NSF types are clearly more dispersed however a roughness surface is appreciated in the polymer. However, this could be related to the correct dispersion of nanoclays, as is observed in the literature [27,43,44], such a huge degree which is clearly increased in the 8%-reinforced nanoclay nanocomposite seems to correspond with polymer degradation. In the case of polar samples with 8% filler, the samples show less appreciable nanoclays, probably due to bigger aggregates.

Thermal stability and main transition temperatures were also analyzed to understand the effect of nanoclay type and content in the processability of the material as it seems to have a huge impact on the properties of the nanocomposites and mainly in NSF types. The results obtained from the TGA are shown in Table 4.

Two different behaviors are observed from the TGA results: the NSF-reinforced materials showed a clear reduction of the starting temperatures, and the nanoclay contents also harm the starting temperature of the degradation. In the T_5%_, the increment of the nanoclay content and the NSF type showed lower temperatures, with a maximum of 24 °C for the PLA+8%NSF material. In T_10%_, a reduction is not noticeable as in the T_5%_, but again the same tendency is observed. Nonetheless, a small shift (4 °C) to higher temperatures is observed for hydrophilic nanoclays. The results are in concordance with the assumption in the previous test of a higher PLA degradation in NSF nanocomposites during the process. Similarities have been observed in the literature with surface-modified and organic reinforcements [32,45]. On the other hand, the T_max_ is increased for all the materials and mainly for NSF nanocomposites. It could be related to the thermal stabilization of nanoclays and stronger interactions with PLA [25]. The residue is increased with the nanoclay content, but always higher in the polar nanoclays due to the higher inorganic content that does not degrade.

Main transition temperatures were also analysed. DSC thermographs are shown in Figure 7.

Main temperatures, enthalpy of the melting, and the cold crystallization and crystallinities are shown in Table 5 for better discussion. The T_g_ shifted in all the materials to higher temperatures due to the presence of nanoclays. However, the lower shifts were obtained for NSF and high contents. The lower shifts of NSF can be related to the degradation observed in TGA and the mechanical properties obtaining lower chains. The effect of the high content is also related to the lower intercalation of the nanoclay layers. The T_c_ of NSF is lower than the observed for hydrophilic nanoclays but enthalpy is higher than the polar ones. It is probably due to the higher dispersion and of them, enhancing the crystallization. Moreover, the lower temperatures could be also associated with the degradation of the PLA. In addition, two different peak behaviors were observed for both types of nanoclays. Hydrophilic nanoclays produced a broader peak while the NSF ones had a narrow peak. The difference in the peak behavior could be also related to the degradation of polymer chains, which could enhance also the crystallinity content. However, the melting behavior is almost the same with slightly higher temperatures for the hydrophilic ones. The enthalpy and crystallinity of the samples during the melting are almost the same for both nanoclays when the same content is used. Thus, a higher crystal content is formed during the cold crystallization of the NSF materials, while it seems that polar can enhance better the crystallization during the cooling process of the materials. The difference can be related to the higher interaction of PLA with NSF that can disturb the crystal growth during the cooling process. Moreover, a negative effect is observed in the crystallinity of all the materials due to the presence of nanoclays.

### 3.3. Mechanical Assessment in a Product

The mechanical assessment of the materials was simulated in a commercial food tray. The tray, as mentioned above, was designed from a real design. The real product and digital mock-up are shown in Figure 8.

The digital mock-up is a simplified version of the real one. The most significant change is that the mock-up did not consider the bottom ribs. These ribs are used to stiffen the tray bottom and were avoided in order to increases the easiness of thin walls meshing and avoid stress concentration phenomena in the rib corners due to the difficulty of meshing small fillet radius. The load was evaluated at 10 kN force and was placed in the inner bottom area of the tray. This load doubles the maximum product content for such trays. This safety factor was introduced as a stress increase factor. All degrees of freedom were limited for the upper area of the tray. This restriction was imposed to study the vertical deformation of the tray and avoid flexion loads in the walls and the upper border. An example of the results for the deformation is shown in Figure 9.

The results of the von Misses tensions, the maximum deformation, and the safety factor are shown in Table 6 for commercial materials and Table 7 for the nanocomposites obtained in this work. As expected, as the loads or the geometry was the same for all tested materials, no changes are observed in the Von Misses tensions, which are concentrated in the creases of the food tray. The maximum deformation is observed in the center of the tray. However, the deformation is reduced when the stiffness of the material is increased. The lowest deformations are obtained for the NSF materials, but also polar nanoclay-reinforced materials showed the same or lower deformation than PLA and the other commercial polymers. The safety factor has to be over 1 to be considered as acceptable material for the product. For the food tray, all the analyzed materials showed higher safety factors than 30. Nonetheless, the high safety factors in PLA nanocomposites, competitive again the commercial ones, means that changes in the design (as a change in the thickness of the tray) can be done with successful results.

Another simulation mimicking the handling of the tray was carried out. The movement degrees of freedom of two opposite borders were limited as such borders are the preview handling areas. (Figure 10). The applied load was 10 kN following the logic of the previous analysis. Allowing the deformation non fixed borders allows flexural loads on the vertical walls. As a consequence, a deformation is observed in the lateral parts of the food tray. Nonetheless, the maximum tensions were the same or increased slightly from the previous one (Table 8 and Table 9). The deformations were slightly increased for all the studied materials, but the same trend was observed. The safety factor was slightly reduced but in all the cases the value was over 30. It must be stated that the model presents a regular wall thickness, while the thermoformed tray has not. Thus, a 30 safety factor can be considered appropriate but does not avoid the necessity for experimental tests.

Figure 11 shows the ratings obtained by commercial polymers used for food packaging and the produced nanocomposites in terms of deformation and safety factors. The blue square marks the ranges defined by the safety factors and deformations obtained with commercial materials, and the yellow square denotes the area defined by nanoclay composites. The safety factor is sufficiently high in all the cases to ensure a satisfactory deployment of the materials as a food tray. The safety factor can be reduced by decreasing the thickness of the tray wall, but this can compromise the ability of the composite films to be thermoformed. Thus, the authors prefer using the same thickness found in the original product. Main differences in the behaviors of the materials were found in their deformations under use. While commercial materials showed deformations below 1 mm, specifically in the range between 0.18 and 0.55 mm, nanoclay composites showed deformations lower than any commercial material. In the case of the tray, such differences can be neglected because are difficult to detect by any user. Thus, nanoclay composites showed a better deformation than commercial materials. Deformation is a milestone in application and design. High deformations or visually recognizable ones are undesired and in the case of packaging, a high deformation can cause a negative visual effect of the packed material. The use of nanocomposites ensures a safety factor comparable to commercial materials and lower deformations which enables its use for the application without any design change.

### 3.4. LCA Preliminary Results

A preliminary LCA was performed using the sustainability pack for the common plastics (HDPE, PP, and PET). The plastic production and use were fixed in Europe with a one-year use of the plastics. The geographical production and use were fixed in Europe where the tax of recyclability, energy recovery, and landfill is known for the last 2019 [5]. Moreover, no recyclable materials were used for the calculus of the manufacture of the injection mold food tray as this use is still incipient in food packaging. The software has no data for plain PLA and nanocomposites but they were recalculated from literature data. PLA global potential warming (GPW) was obtained from LCA Naturworks inventories [46]. The end life of PLA was considered 100% landfill and the data from composting values in the literature were used [47,48]. The GPW associated with transport and product manufacture was initially obtained from literature, however, due to the high differences (also in the case of PET), the values were recalculated from the Solidworks parameter and applying the matrix density. PLA-nanoclay composite calculus was obtained from the combination of the PLA and nanoclay data and related to the phase weight [49]. However, in the case of nanoclays, two assumptions were made: The material value for polar nanoclays is approximately half of the surface-modified nanoclays due to the modification and purification associated with that reaction, highly energetically costly; and there is no value associated to the landfill of these materials.

Figure 12 shows the sustainability of PLA and PLA nanocomposites in comparison to the common plastics used for their manufacture in terms of GWP. The most important reduction in GWP is associated with material production. PLA has a GWP of 0.6 kg CO_2_/ kg of PLA. It is clear the sustainability of the PLA matrix in comparison with PE, PP, and PET, which achieved values of 1.8–1.9, 1.6–1.9, and 2–2.4 kg CO_2_/kg polymer, respectively. The low GWP of PLA is due to the CO_2_ uptake of the raw material used for its production during the growth of the plant. Product manufacture and transport are mainly related to the density and viscosity, and process temperatures of the material, and a slightly higher value is obtained in PET, PLA, and PLA-nanocomposites. The end life value of the studied materials showed significant differences regarding renewable and petrol-based materials. In the case of HDPE, PP, and PET, the end of life is mainly related to the recyclability of the materials and the energy recovery. The burning of the material produces a high quantity of warming gases. The landfill tax has a poor effect on this value as these plastics have a low degradation rate. On the other hand, the values of PLA and PLA nanocomposites are related to the gases produced during the composting of the materials, which generally have higher potential warming values in terms of g CO_2_ equivalents than the CO_2_. Moreover, these gases can be collected and used for energy production. Thus, the higher value does not have to be directly understood as a more unsustainable material at its end of useful life. Moreover, PLA is recyclable which was not considered in the calculus. The recyclability of PLA can reduce this value.

The increment of the nanoclay content in the PLA composites has a small reduction of the GWP values. Nonetheless, the reduction is really small to be considered important and is mainly related to the density values of these materials.

In terms of energy, the same recalculations were done for PLA and PLA-nanocomposites, except the product manufacture as the value is similar to the obtained in the literature and this value was preferred against recalculations. The results are represented in Figure 13.

Similar behavior is observed in terms of energy. The most consuming material was PET, followed by HDPE and PP. PLA materials showed smaller values as the raw material production is really small in comparison with the extraction from petrol. Product manufacture and transport are similar in all the cases as the equipment is supposed to be similar and the effect of the fluid properties and density has a small effect. In the case of transport, the energy is really small that the increment in the weight due to the higher density has a poor effect. End life energy is really small that it is not appreciable in Figure 13. A magnification of the results is shown in Figure 14.

In terms of energy, the end life of the PLA and PLA nanocomposites are the lower values, the lowest being the PLA+8%NSF, but the difference is smaller as observed in the total energy. The high energy consumption of the PLA materials is due to the composting conditions. Moreover, a reduction of the energy is obtained in the petrol-based polymers due to the energy recovering considered in the calculus of the end of life. However, as mentioned above, PLA and PLA nanocomposites were just considered for compositing while they can be recycled. The effect of the recyclability is smaller than the composting process and implying more sustainable materials.

## 4. Conclusions

PLA nanocomposite materials were prepared through a masterbatch methodology to improve the dispersion and intercalation of nanoclays usually obtained by melt procedures while facilitating the manufacturing process. A comparison of both methodologies was examined with the 4% *w*/*w* of polar nanoclay-reinforced PLA. The masterbatch methodology showed similar or slightly better results than the direct compounding of both phases in terms of mechanical and intercalation by XRD and DSC results. Moreover, thermal degradation was not affected by the addition of another step in the processing. Thus, the results confirm the effectiveness of the masterbatch technology in comparison with the direct melt. Moreover, this methodology avoids the continuous feeding of the powder and facilitates obtaining the correct composition.

Composites materials with 2 to 8% *w*/*w* of polar and surface modified nanoclays were prepared with the masterbatch technology in order to assess the impact of the filler content and polarity. Higher resistance and deformations were obtained with polar nanoclays while a better modulus was observed in NSF. Lower resistance and deformation in NSF, although a better dispersion was observed, were associated with degradation of the matrix due to the organomodifiers of the nanoclays. Some degree of intercalated structure is observed in polar nanoclays at low contents and the thermal stability was not affected.

Finally, the mechanical assessment and sustainability were studied and compared with common plastics with the simulation of a food tray. All the composites materials showed a better performance in the tray than common plastics and the PLA itself. The preliminary LCA carried out also showed the sustainability of the PLA nanocomposites, even when recycling was not considered in the studied nanocomposites.

## Figures and Tables

**Figure 1 polymers-13-02133-f001:**
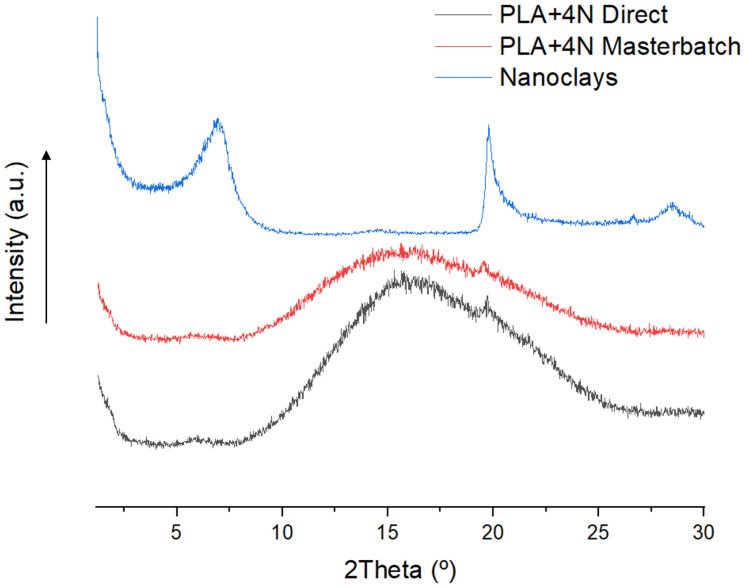
XRD diffractograph of PLA+4%N nanocomposite with the different methodologies and the nanoclays used.

**Figure 2 polymers-13-02133-f002:**
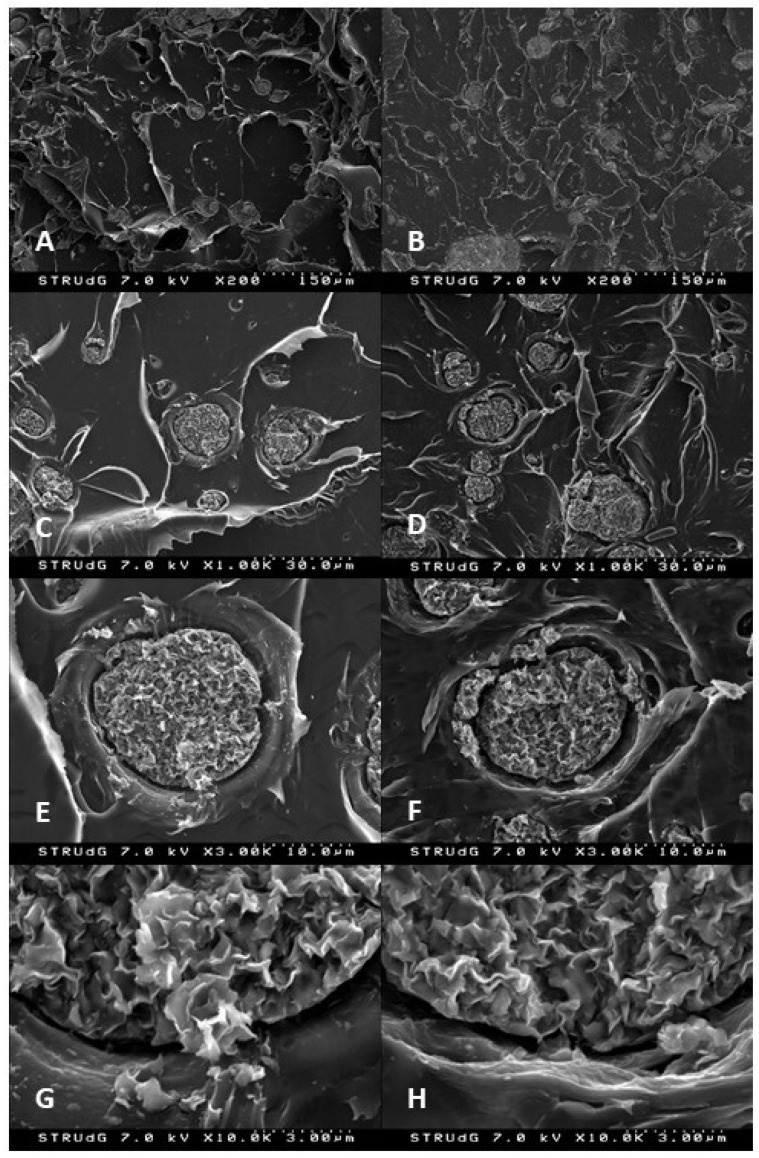
SEM micrographs of PLA+4%N nancomposites prepared with direct melting (left column) and masterbatch methodology (right column) at different magnifications: (**A**,**B**) by ×200; (**C**,**D**) by ×1000, (**E**,**F**) by ×3000; and (**G**,**H**) by ×10,000.

**Figure 3 polymers-13-02133-f003:**
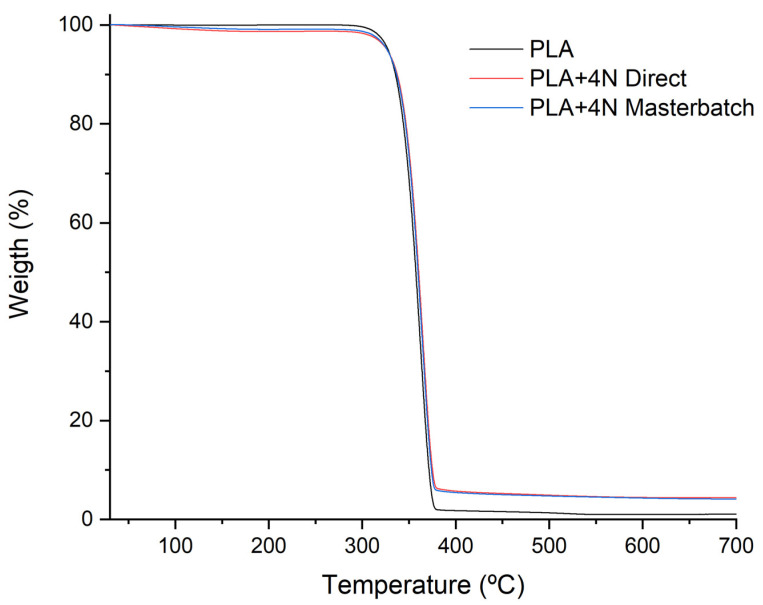
TGA termograph of PLA and PLA+4%N composites.

**Figure 4 polymers-13-02133-f004:**
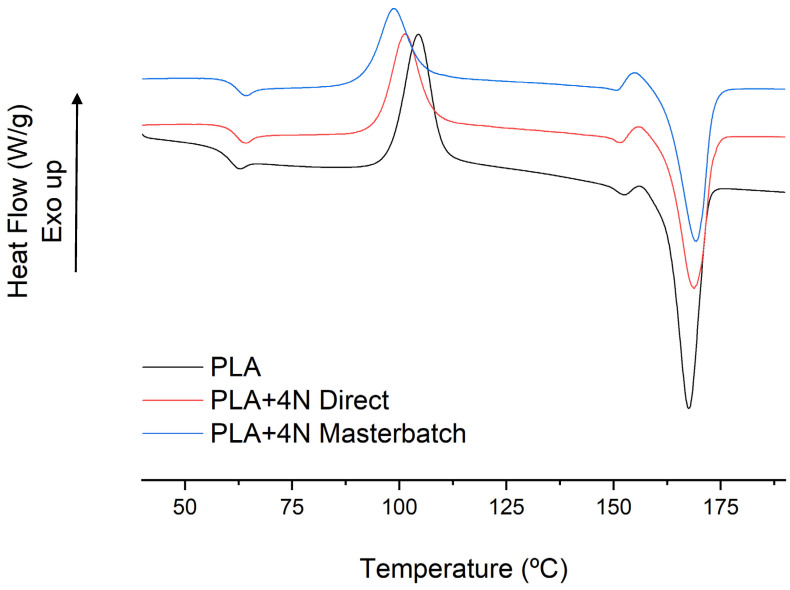
DSC thermograph of PLA and PLA nanocomposites produced with both methodologies.

**Figure 5 polymers-13-02133-f005:**
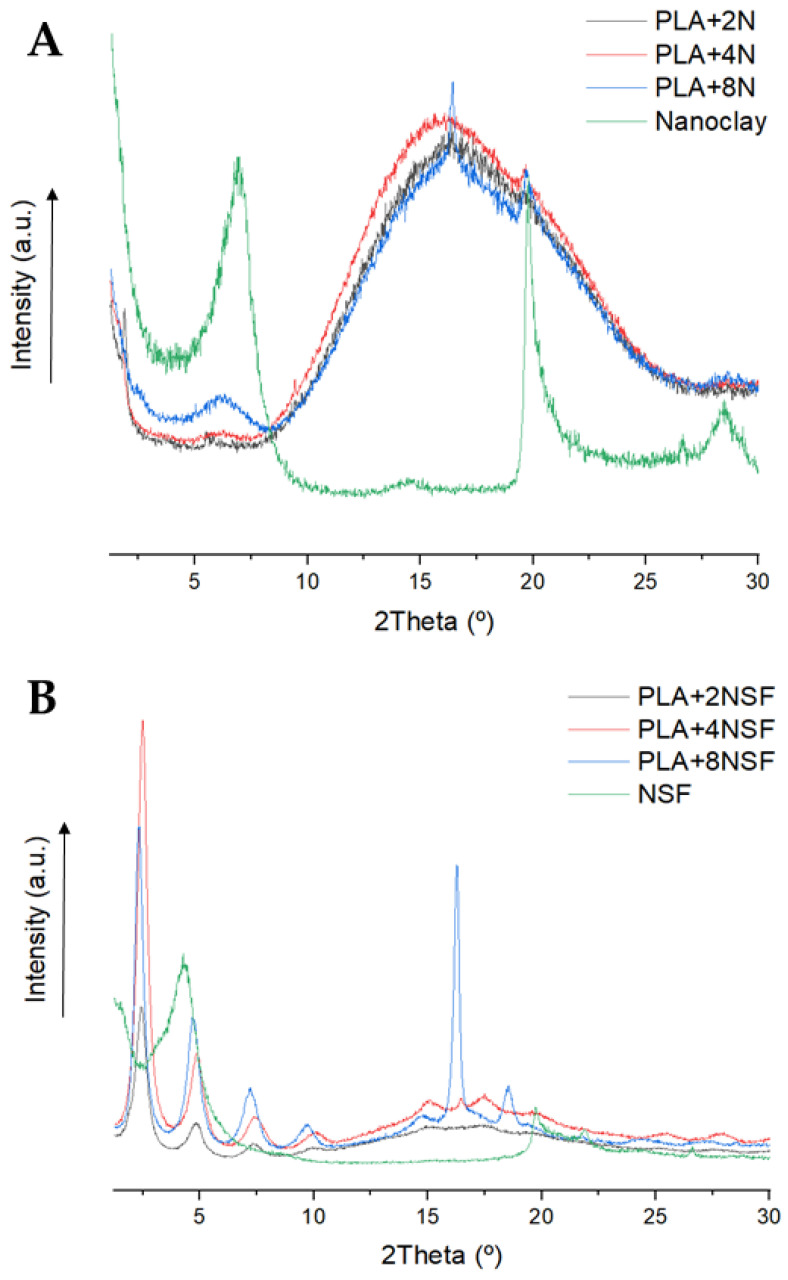
XRD patterns of all the analyzed composites. (**A**) N type composite (**B**) NSF-reinforced composites.

**Figure 6 polymers-13-02133-f006:**
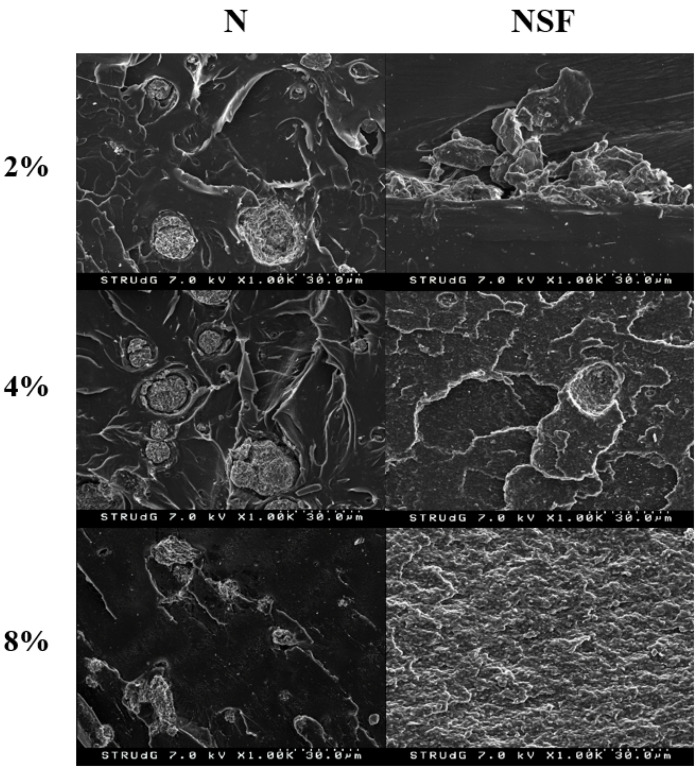
SEM micrographs PLA nanocomposites regarding the reinforcement content and the nanoclay type.

**Figure 7 polymers-13-02133-f007:**
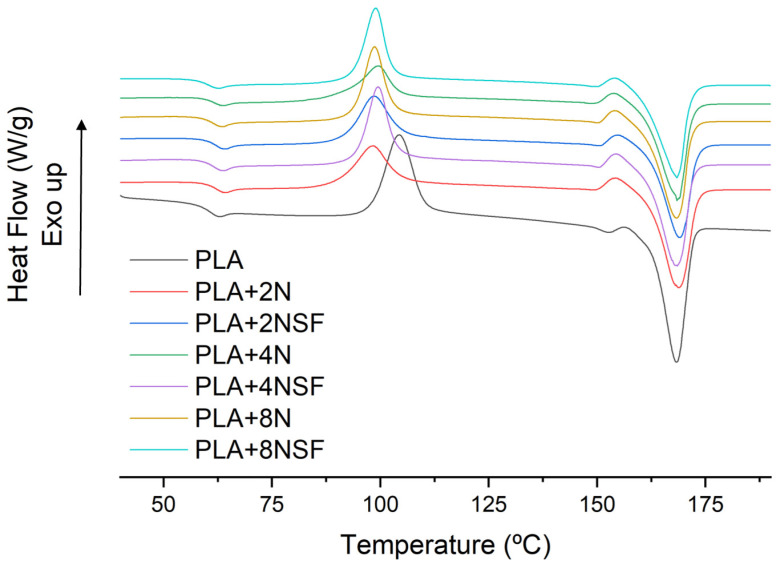
DSC thermographs of the second heating of all the studied materials.

**Figure 8 polymers-13-02133-f008:**
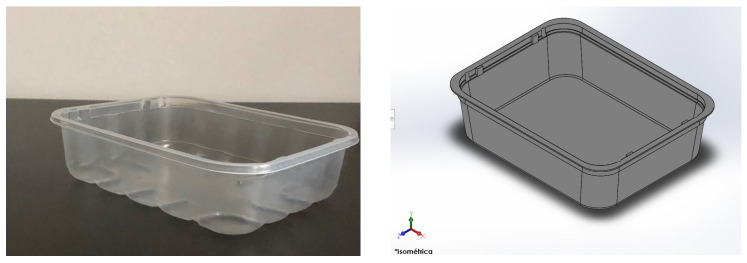
Original food tray and digital mock-up prepared from it.

**Figure 9 polymers-13-02133-f009:**
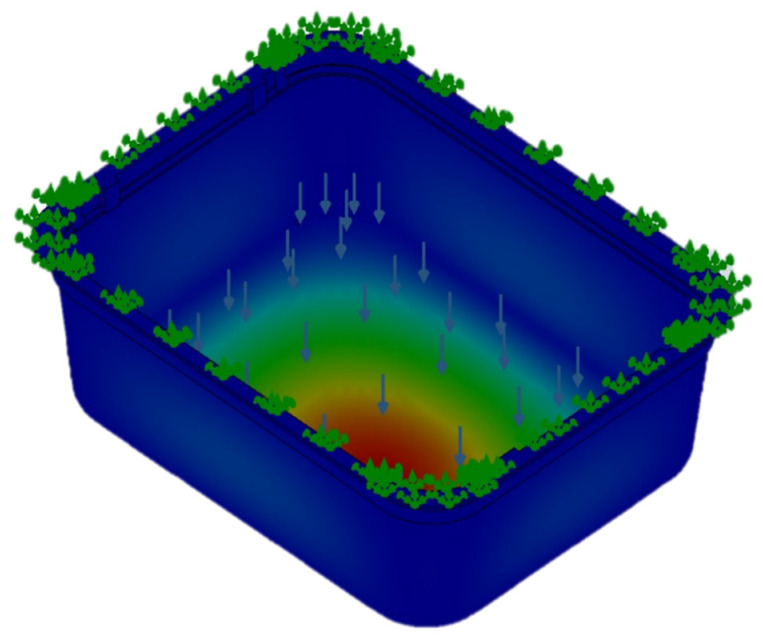
Example of the deformation results of the simulation in the food tray.

**Figure 10 polymers-13-02133-f010:**
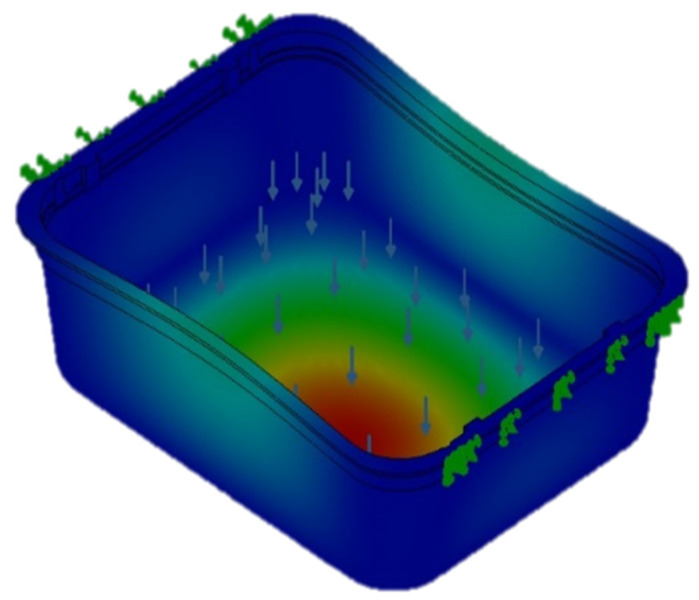
Example of the deformation results for the second simulation of the food tray.

**Figure 11 polymers-13-02133-f011:**
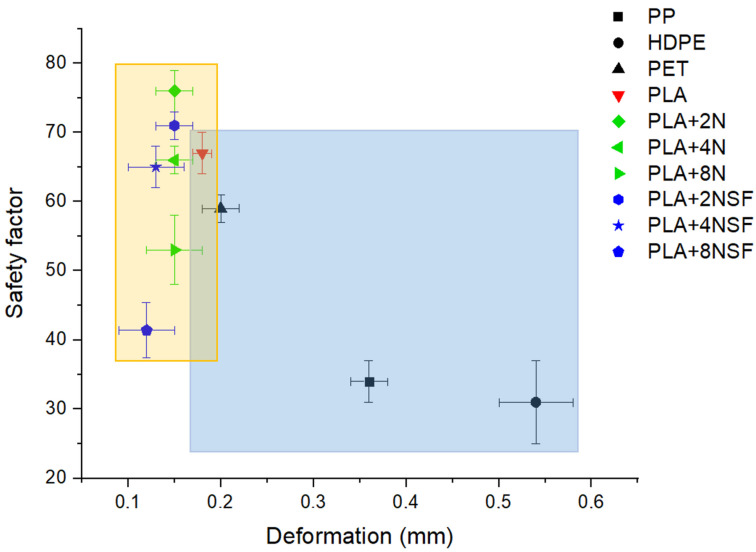
Comparison between commercial polymers and produced nanocomposites in terms of deformation and safety factors. The blue square marks the results for common polymers, including PLA, while the orange indicates the composites samples.

**Figure 12 polymers-13-02133-f012:**
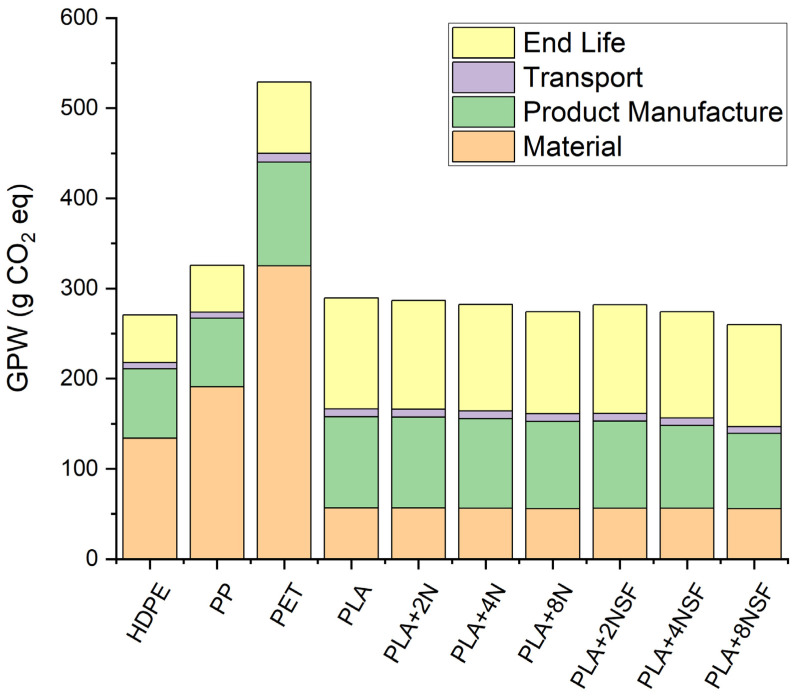
LCA results for analyzed materials in terms of CO_2_ equation.

**Figure 13 polymers-13-02133-f013:**
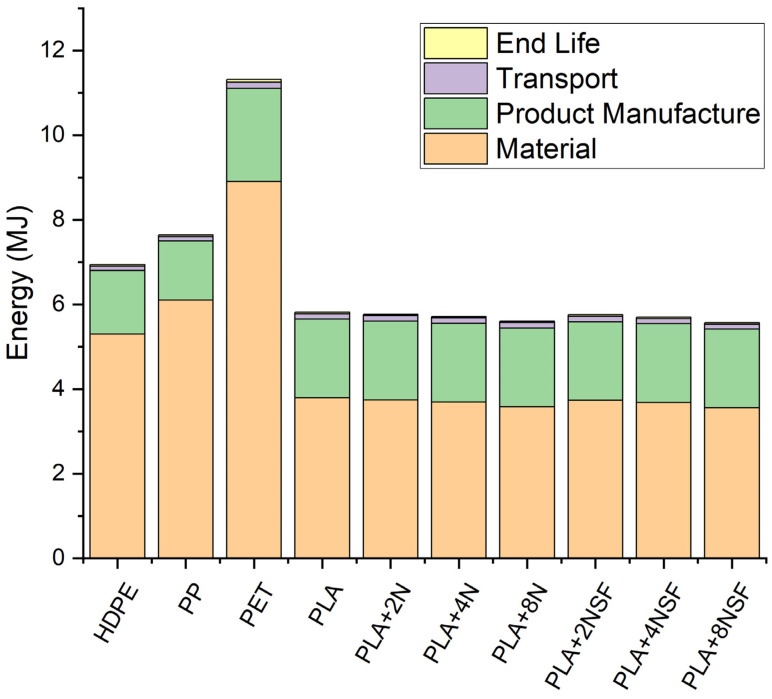
LCA results in terms of energy.

**Figure 14 polymers-13-02133-f014:**
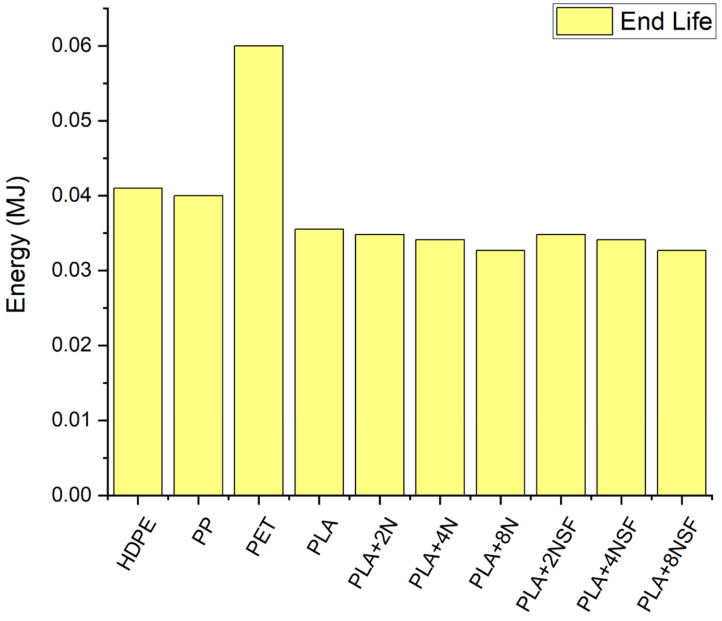
LCA results for end life in terms of energy.

**Table 1 polymers-13-02133-t001:** Tensile and Flexural Properties of PLA and PLA+4%Nanoclays using the different compounding methods.

Sample	Young’s Modulus (GPa)	Tensile Strength (MPa)	Tensile Deformation at Maximum Force (%)
PLA	3.34 ± 0.13	49.9 ± 1.3	2.3 ± 0.1
PLA+4%N Direct	3.94 ± 0.05	47.0 ± 3.4	2.3 ± 0.3
PLA+4%N Masterbatch	4.03 ± 0.11	49.0 ± 1.1	2.5 ± 0.1

**Table 2 polymers-13-02133-t002:** Degradation temperatures for PLA+4%N nanocomposites and neat PLA.

Temperatures (°C)	PLA	PLA+4%N Direct	PLA+4%N Masterbatch
T5%	328	326	327
T10%	334	337	337
Tmax	362	365	364
Residue (700 °C) (%)	1.04	4.38	4.12

**Table 3 polymers-13-02133-t003:** Mechanical properties of PLA and PLA nanocomposites.

Sample	Young’s Modulus (GPa)	Tensile Strength (MPa)	Deformation at Maximum Force (%)
PLA	3.34 ± 0.13	49.9 ± 1.3	2.3 ± 0.1
PLA+2%N	4.00 ± 0.16	56.5 ± 0.7	2.8 ± 0.2
PLA+4%N	4.03 ± 0.11	49.0 ± 1.1	2.5 ± 0.1
PLA+8%N	4.18 ± 0.07	36.1 ± 3.0	1.5 ± 0.2
PLA+2%NSF	4.19 ± 0.12	52.8 ± 1.5	2.5 ± 0.2
PLA+4%NSF	4.56 ± 0.19	48.9 ± 2.8	2.2 ± 0.2
PLA+8%NSF	5.01 ± 0.27	28.2 ± 1.8	1.3 ± 0.2

**Table 4 polymers-13-02133-t004:** Degradation temperatures and residue at 700 °C of PLA and PLA nanocomposites.

	T5% (°C)	T10% (°C)	Tmax (°C)	Residue at 700 °C (%)
PLA	328	334	362	1.04
PLA+2%N	328	338	367	2.28
PLA+4%N	327	337	364	4.12
PLA+8%N	318	332	365	6.57
PLA+2%NSF	323	334	364	0.71
PLA+4%NSF	320	333	369	2.78
PLA+8%NSF	304	326	367	5.33

**Table 5 polymers-13-02133-t005:** Main transition temperatures, cold crystallization and melting enthalpy, and crystallinity of the samples.

	T_g_ (°C)	T_c_ (°C)	ΔH_c_ (J/g PLA)	T_m_ (°C)	ΔH_m_ (J/g PLA)	Crystallinity (%)
PLA	57.7–60.8	104.3	38.7	167.5	50.1	53.6
PLA+2%N	59.7–62.4	98.3	21.5	168.7	39.4	42.1
PLA+4%N	60.3–62.4	98.7	22.6	169.3	37.8	40.3
PLA+8%N	59.5–62.2	99.4	18.4	168.4	40.3	43.1
PLA+2%NSF	59.2–62.2	99.4	26.5	168.5	39.2	41.9
PLA+4%NSF	59.1–61.6	98.5	24.5	168.4	38.3	40.9
PLA+8%NSF	57.9–61.0	98.9	26.5	168.4	41.1	44.0

**Table 6 polymers-13-02133-t006:** Simulation results for commercial materials.

	PP	HDPE	PET	PLA
Von Misses (MPa)	0.7 ± 0.1	0.7 ± 0.1	0.7 ± 0.1	0.7 ± 0.1
Maximum deformation (mm)	0.32 ± 0.02	0.48 ± 0.05	0.18 ± 0.02	0.16 ± 0.01
Safety factor	35 ± 2	32 ± 5	56 ± 2	69.0 ± 2

**Table 7 polymers-13-02133-t007:** Simulation results for the nanocomposites.

	PLA+2%N	PLA+4%N	PLA+8%N	PLA+2%NSF	PLA+4%NSF	PLA+8%NSF
Von Misses (MPa)	0.7 ± 0.1	0.7 ± 0.1	0.7 ± 0.1	0.7 ± 0.1	0.7 ± 0.1	0.7 ± 0.1
Maximum deformation (mm)	0.16 ± 0.01	0.14 ± 0.01	0.13 ± 0.03	0.13 ± 0.02	0.12 ± 0.02	0.11 ± 0.03
Safety factor	78 ± 2	68 ± 2	55 ± 4	73 ± 2	68 ± 2	43 ± 3

**Table 8 polymers-13-02133-t008:** Results for the second simulation for commercial polymers.

	PP	HDPE	PET	PLA
Von Misses (MPa)	0.7 ± 0.1	0.7 ± 0.1	0.7 ± 0.1	0.8 ± 0.1
Maximum deformation (mm)	0.36 ± 0.02	0.54 ± 0.04	0.20 ± 0.02	0.18 ± 0.01
Safety factor	34 ± 3	31 ± 6	59 ± 2	67 ± 3

**Table 9 polymers-13-02133-t009:** Results for the second simulation using the new nanocomposites.

	PLA+2%N	PLA+4%N	PLA+8%N	PLA+2%NSF	PLA+4%NSF	PLA+8%NSF
Von Misses (MPa)	0.8 ± 0.1	0.8 ± 0.1	0.8 ± 0.1	0.8 ± 0.1	0.8 ± 0.1	0.8 ± 0.1
Maximum deformation (mm)	0.15 ± 0.02	0.15 ± 0.02	0.15 ± 0.03	0.15 ± 0.02	0.13 ± 0.03	0.12 ± 0.03
Safety factor	76 ± 3	66 ± 2	53 ± 5	71 ± 2	65 ± 3	41 ± 4

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
