# Peer review of "Nanocomposites Materials of PLA Reinforced with Nanoclays Using a Masterbatch Technology: A Study of the Mechanical Performance and Its Sustainability"

_polymers, 2021, doi:10.3390/polym13132133_

Round 1

Reviewer 1 Report

Comments on polymers-1243262

In this paper, PLA nanocomposites were prepared through a masterbatch methodology, which improve the dispersion and exfoliation of nanoclays. Higher resistance and deformations were obtained with polar nanoclays. In addition, the mechanical assessment and sustainability of PLA nanocomposites were studied with the simulation of a food tray and showed a better performance. However, there are still some issues to be addressed before it can be possibly accepted for publication.

  1. The language needs to be polished to make the paper better read and understand. In addition, the font in the table should be unified.
  2. In section 3.1,

-the tensile and flexural fracture surface of PLA nanocomposites should be observed to explain the mechanical properties.

-The XRD results of nanoclay should also be given.

-The author said that “The increment in the distance indicates an intercalated structure in the nanocomposite”, the mechanism for the exfoliation of nanoclay should be given.

  1. In section 3.2, since many test results are compared in terms of nanoclay type and content, such as mechanical properties, XRD, TGA and DCS, what is the main conclusion of this part? The content in this part is too lengthy.
  2. In figure 9, the colors represented for commercial polymers and produced nanocomposites should be more different.
  3. Some highly relevant literature on mechanical properties of PLA should be cited, such as Compos. Part A, 2021,144, 106317; Biomacromolecules 22 (4), 1432; ACS Sustain Chem Eng 2020; 8(44):16612; Chem Eng J 2020;397:125336; Part B: Eng, 2020, 190, 107930; ACS Omega, 2018, 3, 5615; and ACS Sustainable Chem. Eng. 2017, 5 (9), 7894

Author Response

In this paper, PLA nanocomposites were prepared through a masterbatch methodology, which improve the dispersion and exfoliation of nanoclays. Higher resistance and deformations were obtained with polar nanoclays. In addition, the mechanical assessment and sustainability of PLA nanocomposites were studied with the simulation of a food tray and showed a better performance. However, there are still some issues to be addressed before it can be possibly accepted for publication.

  1. The language needs to be polished to make the paper better read and understand. In addition, the font in the table should be unified.

An exhaustive revision of the manuscript has been done in order to facilitate the read and the understand of the paper. Table font has been unified.

  1. In section 3.1,

-the tensile and flexural fracture surface of PLA nanocomposites should be observed to explain the mechanical properties.

The authors agree with the reviewer and SEM pictures from tensile fracture surface has been included on the manuscript.

-The XRD results of nanoclay should also be given.

The XRD results of nanoclay were not performed as they could not be done with the XRD equipment. The sampler is not available in our installations. Besides, nanoclays has been largely studied in the literature and some cites has been added in order to reinforce the differences in the results analysed

-The author said that “The increment in the distance indicates an intercalated

structure in the nanocomposite”, the mechanism for the exfoliation of nanoclay should be given.

The exfoliation mechanism produced during the mixing it’s difficult to be determinate by the results obtained. Authors believe that the increment in the plane distance of the nanoclays is a consequence of the presence of some intercalation in the nanoclays plates by the presence of the polymer. This effect has been observed in the literature as it is cited in the article.

In section 3.2, since many test results are compared in terms of nanoclay type and content, such as mechanical properties, XRD, TGA and DCS, what is the main conclusion of this part? The content in this part is too lengthy.

The main conclusions of such part is the effectiveness of polar nanoclays in comparison with organic surface modified nanoclays. Organic modified nanoclays showed a better performance in comparison with non-modified due the better affinity with the polymer. Nonetheless, in these work it is shown that adequate dispersion and some degree of intercalation is obtained for polar nanoclays although the differences in polarity. Besides, organic modified nanoclays reduce the thermal stability of the nanocomposites materials.

The discussion has been modified to highlight these conclusions and facilitate the reading.

  1. In figure 9, the colors represented for commercial polymers and produced nanocomposites should be more different.

The authors agree with the reviewer and the figure has been modifies to easily differentiate the commercial polymers from the nanocomposites.

  1. Some highly relevant literature on mechanical properties of PLA should be cited, such as Compos. Part A, 2021,144, 106317; Biomacromolecules 22 (4), 1432; ACS Sustain Chem Eng 2020; 8(44):16612; Chem Eng J 2020;397:125336; Part B: Eng, 2020, 190, 107930; ACS Omega, 2018, 3, 5615; and ACS Sustainable Chem. Eng. 2017, 5 (9), 7894

The authors thank the reviewer for the interesting literature, however the articles were not clearly related with article topic as them are devoted mainly to fire retardant particles. The article which could be used to justify some effect of nanoparticles reinforcements have been included in the document.

Reviewer 2 Report

The paper “Nanocomposites materials of PLA reinforced with nanoclays using a Masterbatch technology: A study of the mechanical performance and its sustainability” investigated a premixing procedure, technically named masterbatch. The manuscript is well organized; however, to improve the quality, the following recommendations can be incorporated.

1.The authors should review the other more new investigation on their study way in the introduction part and finally note the novelty of the article. The introduction part needs to develop.

  1. Authors can cite the following work in the introduction which is closely related to their work and recently reported:

-"Experimental Investigation of Polymer Solutions Used in Enhanced Oil Recovery-Thermal properties Improved by Nanoclay." In 77th EAGE Conference and Exhibition 2015, vol. 2015, no. 1, pp. 1-3. European Association of Geoscientists & Engineers, 2015./ "Biodegradation assessment of poly (lactic acid) filled with functionalized titania nanoparticles (PLA/TiO 2) under compost conditions." Nanoscale research letters 14, no. 1 (2019): 1-10.

3- Explain the results obtained from the monitoring of simulation analysis more thoroughly. Complete information about the simulator and the simulation steps is required.

4- Technical terms are misused through the manuscript and the writing needs a revision.

5- Need to provide characterization tests for the materials used (such as SEM or TEM or ...).

6- Most of the references are old, new articles should be used as much as possible.

7- Results of LCA needs to error and standard division.

Author Response

The paper “Nanocomposites materials of PLA reinforced with nanoclays using a Masterbatch technology: A study of the mechanical performance and its sustainability” investigated a premixing procedure, technically named masterbatch. The manuscript is well organized; however, to improve the quality, the following recommendations can be incorporated.

The authors want to thank the kindly reviewer comments and would do the effort to incorporate all it is possible to improve the paper. 

  1. The authors should review the other more new investigation on their study way in the introduction part and finally note the novelty of the article. The introduction part needs to develop.

A review of all the document, but putting and effort on the introduction has been done to highlight the novelty of the article. All the changes are marked with the Microsoft track engine.

  1. Authors can cite the following work in the introduction which is closely related to their work and recently reported:

-"Experimental Investigation of Polymer Solutions Used in Enhanced Oil Recovery-Thermal properties Improved by Nanoclay." In 77th EAGE Conference and Exhibition 2015, vol. 2015, no. 1, pp. 1-3. European Association of Geoscientists & Engineers, 2015./ "Biodegradation assessment of poly (lactic acid) filled with functionalized titania nanoparticles (PLA/TiO 2) under compost conditions." Nanoscale research letters 14, no. 1 (2019): 1-10.

The author thanks the articles proposed by the reviewer and have been included in the article.

3- Explain the results obtained from the monitoring of simulation analysis more thoroughly. Complete information about the simulator and the simulation steps is required.

The authors have added the folloving information to methods section, in order to complete the information of the analysis:

The finite element analysis was performed in the same software with the advanced simulation analysis package. The simulation included the usual steps for such analysis; definition of the material properties (Young’s modulus, Poison’s ratio and tensile strength), placing movement restrictions and loads, meshing the model, launching the analysis and review of the results

4- Technical terms are misused through the manuscript and the writing needs a revision.

The authors has done an exhaustive revision of the documents. The authors hope that the errors has been corrected.

5- Need to provide characterization tests for the materials used (such as SEM or TEM or ...).

The authors agree with the reviewer that TEM will be quite useful. Sadly, the TEM equipment from the University has been on reparation long time. SEM pictures has been taking and the results included and commented in the article.

6- Most of the references are old, new articles should be used as much as possible.

New references from recent articles has been included in the manuscript. Nonetheless, the authors preferred to also keep the oldest one as much as possible.

7- Results of LCA needs to error and standard division.

Errors from the LCA cannot be computed as the data used for the analysis didn’t include them. The LCA preliminary analysis has been performed based on the data obtained from literature and the software.

Round 2

Reviewer 1 Report

The authors have addressed some of my concerns, but there are still some issues which have been been addressed properly. So I cannot recommend it to be published at this stage.

  1. Regarding the XRD results of nanoclay should also be given,

The XRD pattern of nanoclay should be performed by XRD. The authors' statement is not right, and in fact  there is a sample holder for solid powder for XRD testing. Though nanoclays have been widely characterized by XRD in the literature, the nanoclay may show different interlayer distance depending on the type of intercalation agents.

2. With respect to the suggested references, there are some relevant to DSC, thermal stability and mechanical properties in addition to fire retardancy. I strongly encourage authors to cite. 

Author Response

The authors have addressed some of my concerns, but there are still some issues which have been addressed properly. So I cannot recommend it to be published at this stage.

  1. Regarding the XRD results of nanoclay should also be given,

The XRD pattern of nanoclay should be performed by XRD. The authors' statement is not right, and in fact there is a sample holder for solid powder for XRD testing. Though nanoclays have been widely characterized by XRD in the literature, the nanoclay may show different interlayer distance depending on the type of intercalation agents.

The authors stated that our university equipment couldn’t perform the powder testing. The sample were send to another university and the results has been added to the text. 

  1. With respect to the suggested references, there are some relevant to DSC, thermal stability and mechanical properties in addition to fire retardancy. I strongly encourage authors to cite. 

The authors cited all the literature proposed by the reviewer in the previous review as recommended.

Reviewer 2 Report

The authors considered the comments of the reviewer. The revised manuscript is significantly improved. However, because some parts are still obscure, illustrate better the "comparison between the commercial polymers used for food pack aging and the produced nancomposites in terms of deformation and safety factors" and the results obtained figure 11.

The authors should indicate experimental errors throughout the paper.

Author Response

Reviewer 2

The authors considered the comments of the reviewer. The revised manuscript is significantly improved. However, because some parts are still obscure, illustrate better the "comparison between the commercial polymers used for food pack aging and the produced nanocomposites in terms of deformation and safety factors" and the results obtained figure 11.

The authors should indicate experimental errors throughout the paper.

The authors thanks the reviewer for the comment and the comparison results has been enlarged. Regarding the experimental errors, thermal and structural properties were performed in a unique test which no error could be obtained. The theoretical errors of the simulations were obtained from repeating the simulation with the results of the different tensile samples and included in the text.

Round 3

Reviewer 1 Report

The work can be accepted as it is now.